# A Neural-Network-Based Approach to Chinese–Uyghur Organization Name Translation

**Aishan Wumaier** [1,2,†]🆔**, Cuiyun Xu** [1,2,†]**, Zaokere Kadeer** [1,2,*]**, Wenqi Liu** [2,3]**, Yingbo Wang** [4]**, Xireaili Haierla** [1,2]**, Maihemuti Maimaiti** [1,2]**, ShengWei Tian** [2,3] **and Alimu Saimaiti** [3]

[1] School of Information Science and Engineering, Xinjiang University, Urumqi 830046, Xinjiang, China; Hasan1479@xju.edu.cn (A.W.); xucuiyun2020@163.com (C.X.); xireaili25@163.com (X.H.); mahmutjan@xju.edu.cn (M.M.)

[2] Key Laboratory of Multilingual Information Technology in Xinjiang Uyghur Autonomous Region, Urumqi 830046, Xinjiang, China; liuwenqi_2020@163.com (W.L.); tianshengwei@163.com (S.T.)

[3] School of Software, Xinjiang University, Urumqi 830091, Xinjiang, China; almu@iflytek.com

[4] Urumqi Campus, Engineering University of PAP, Urumqi 830049, Xinjiang, China; wyb_pap@163.com

* Correspondence: zuhra@xju.edu.cn; Tel.: +86-186-9015-0614

† These authors contributed equally to this work.

**Abstract:** The recognition and translation of organization names (ONs) is challenging due to the complex structures and high variability involved. ONs consist not only of common generic words but also names, rare words, abbreviations and business and industry jargon. ONs are a sub-class of named entity (NE) phrases, which convey key information in text. As such, the correct translation of ONs is critical for machine translation and cross-lingual information retrieval. The existing Chinese–Uyghur neural machine translation systems have performed poorly when applied to ON translation tasks. As there are no publicly available Chinese–Uyghur ON translation corpora, an ON translation corpus is developed here, which includes 191,641 ON translation pairs. A word segmentation approach involving characterization, tagged characterization, byte pair encoding (BPE) and syllabification is proposed here for ON translation tasks. A recurrent neural network (RNN) attention framework and transformer are adapted here for ON translation tasks with different sequence granularities. The experimental results indicate that the transformer model not only outperforms the RNN attention model but also benefits from the proposed word segmentation approach. In addition, a Chinese–Uyghur ON translation system is developed here to automatically generate new translation pairs. This work significantly improves Chinese–Uyghur ON translation and can be applied to improve Chinese–Uyghur machine translation and cross-lingual information retrieval. It can also easily be extended to other agglutinative languages.

**Keywords:** named entity translation; organization name translation; word segmentation; tagged characterization; syllabification; transformer

## 1. Introduction

In recent years, neural-network-based machine translation has made continual progress for high-resource languages. While the quality of translation has improved significantly, some limitations remain. Previous studies have suggested that translation errors can be divided into several categories, including incorrect word interpretations, grammatical errors, missing translations, named entity translation errors and word order errors [1]. Named entities (NEs) describe key information in text, the incorrect translation of which can be problematic. Thus, the primary goal of machine translation models is to preserve semantic meaning to the greatest possible extent while following the syntactic

structure of the target language. As training texts include both semantic translation and transliteration data, neural networks can be trained using large quantities of parallel sentence pairs. The resulting models may learn specific translation rules from the named entities included in the texts, thereby developing a capacity for translating NEs, even if the NE translation model is not implemented separately [2]. However, low-resource texts typically include a scarce bilingual corpus, exhibiting smaller NE quantities and a lower usage frequency. As a result, the translation performance of low-resource machine translation systems is typically lower than that of comparable high-resource systems [3].

NE translation is a combination of meaning inference and phoneme transliteration. Transliteration is the translation of source language words into a target language, typically utilizing phonetic correspondence. It is often used to translate the names of people, locations, organizations and products into meaningful words in foreign languages [3]. Such names comprise more than 75% of unseen words and pose a challenging problem for machine translation, multilingual information retrieval and other natural language processing applications [4]. Recent studies on the translation results from the LDC2005T34 (the catalog number of the Linguistic Data Consortium Chinese–English Name Entity Lists Version 1.0) large NE corpus have indicated the proportions of successfully transliterated names, place names and organization names to be 100%, 89.4% and 12.6%, respectively [5]. This suggests that the names of people and places can often be transliterated correctly, while most organization names cannot be translated using simple transliteration. It is evident, from these results, that the NE translation quality depends heavily on the entity type, which is likely a result of the complex structure and longer lengths that are common characteristics of organization names, as well as the differences between transliteration and semantic translation.

The Uyghur–Chinese dataset from the 15th China Conference on Machine Translation (CCMT) contains ~170,000 sentence pairs, which contain 6318 organization names. The 1000 development set sentence pairs contain 143 organization names. While there are only a few organization names in the training and testing sentence pairs, a practical Chinese–Uyghur machine translation system needs to translate many more organization names, for which the present translation accuracy is only 15%.

The contributions of our study are as follows: (1) As there are no publicly available Chinese–Uyghur organization name (ON) translation pair corpora, a translation corpus is developed as part of the study. (2) This study presents a neural-network-based translation approach using language-aware input granularity to reduce the influence of language differences and to enhance the ability of machines to learn language-specific features. (3) We apply a recurrent neural network (RNN) attention model and a transformer to Chinese–Uyghur organization name translation tasks with different input granularities, in order to fully explore the performance of the two very different mainstream machine translation architectures. (4) A series of experiments are conducted, the results of which show that Chinese–Uyghur organization name translation benefits from the state-of-the-art neural machine translation framework presented here and is also greatly improved through appropriate word segmentation. (5) In addition, a translation pair generation system is developed to enlarge the corpus automatically.

## 2. Related Works

Named entity research involves both transliteration and translation, although recent studies have focused primarily on machine transliteration. These efforts can be divided into three categories: grapheme, phoneme and hybrid methods [2]. Grapheme-based models [6] use orthographic mapping, while phoneme-based techniques [7] utilize phonetic alignment for transliteration. Hybrid methods apply a combination of several approaches, as well as language-specific knowledge. Neural network (NN) models have recently been explored with varying success, depending heavily on the quantity of available training data. Transliteration has been shown to not perform well when applied to ON translation tasks. This is primarily due to the complexity of ON text, which includes the names of people, places and substantive words. The existing research on NE translation can be divided into

three categories—rule-based [8], statistical [9,10] and NN-based models [11,12], all of which depend heavily on linguistic dictionaries.

These models have been applied to a variety of languages and text structures. For example, Li et al. proposed a conditional random field (CRF) algorithm, combined with a Moses statistical model, for translating Japanese names [13]. Wang et al. proposed a name recognition technique, utilizing a rule-based approach to compensate for a limited corpus and to extract bilingual names for dictionary enhancement [14]. Zhang et al. developed a phrase-based context-dependent joint probability algorithm for NE translation, which consisted of lexical alignment and permutation models [15]. This method was used to extract entities belonging to the organization, industry organization, press and international organization categories. A subset corpus was also isolated for NE translation, containing 135,351 Chinese–English entity pairs and 149,641 English–Chinese entity pairs. The translation accuracy for the former reached 51.5%, with a bilingual evaluation understudy (BLEU) score of 56.1, while the accuracy for the latter reached 36.1%, with a BLEU score of 54.1 [15]. Yang et al. developed a model for Chinese–Slavic Mongolian NE translation based on word alignment, which extracted word pairs automatically [16]. These pairs were then identified using the Chinese NE to produce a Slavic Mongolian candidate translation from a word alignment matrix. Finally, the candidates were filtered using a maximum entropy function.

NE translation can also be approached as a sequence-to-sequence modeling problem. For example, Li et al. introduced a character-level sequence-to-sequence NE translation algorithm that improved machine translation quality by significantly reducing rare words appearing as NEs [11]. Li et al. proposed a sub-word-level English–Chinese person name sequence-to-sequence NE translation algorithm [12]. Yan et al. studied a mainstream structure that incorporated an external NE model into neural machine translation [17]. Kundu et al. proposed both character-based and byte pair-based approaches for neural-network-based methods for NE translation, adopting the recurrent neural network (RNN) encoder–decoder framework and the convolutional neural machine translation (NMT) framework [18]. Najafi et al. presented a comparative study on the NEWS 2018 Shared Task and several diverse systems. Their experiment results showed that a combination of different models obtained the best results [19].

In contrast to several high-resource languages, little has been reported for Chinese–Uyghur alignment or translation due to a lack of available bilingual corpora or NE recognition tools. For instance, Yimamuaishan et al. implemented a rule-based Uyghur–Chinese person name transliteration model [20]. Li et al. introduced a methodology for recognizing and translating Chinese names in Uyghur [21]. Ayiguli et al. investigated the recognition and translation of Chinese and Uyghur time numbers and quantifiers [22]. Wang et al. designed and implemented a Chinese–Uyghur machine translation system for personal resumes based on a combination of templates and dictionaries [23]. Lei et al. applied finite automata combined trigger words with Uyghur NE recognition and translation [24]. This model was used for basic number recognition and translation, as well as to develop a rule-based place name identification algorithm. ON recognition was also conducted using state transitions and keyword matching in order to develop a template-based translation system. However, to the best of our knowledge, no studies on Chinese–Uyghur ON translation currently exist in the literature.

## 3. Methodology

In this section, the Chinese–Uyghur organization name translation task is introduced and some examples are provided. For clarity, Chinese characters and the Chinese phonetic alphabet (CPA) are used together to better describe the Chinese examples, while the Latin script of Uyghur is used for the Uyghur examples. The language-aware word segmentation units are introduced below and the proposed language-aware ON translation approach is used to describe the adopted NMT models. The Chinese–Uyghur ON translation pair generation system architecture is also presented, which was developed using bidirectional ON translation models.

### 3.1. Chinese–Uyghur Organization Name Translation

Organization names typically refer to institutions, such as schools, companies and research or government agencies. These ONs may include place names, brand names, nouns, surname combinations, business terms, industry identification words, technical jargon or abbreviations. New words also frequently appear in organization names and are often not translated correctly. For example, the ONs shown in Table 1 include terms, such as "kashi, black fungus, planting, culture, sports, radio, television and tourism," as well as industry phrasing, like "university, cooperative and bureau." Some ONs are lengthy and are shortened in practical usage. For example, "中科院计算所" (CPA: "Zhōng kē yuàn jì suàn suǒ," which means "Institute of Computing Technology of the Chinese Academy of Sciences") is a common abbreviation for "中国科学院计算技术研究所" (CPA: "zhōng guó kē xué yuàn jì suàn jì shù yán jiū suǒ," which means "Institute of Computing Technology of the Chinese Academy of Sciences").

**Table 1.** Examples of Chinese and Uyghur organization names. CPA: Chinese phonetic alphabet.

| Chinese (CPA) | Uyghur (Latin Script) | English |
| --- | --- | --- |
| 在新疆大学 (zài xīn jiāng dà xué) | Shinjang dashödE | At Xinjiang University |
| 从新疆大学 (cóng xīn jiāng dà xué) | Shinjang dashödin | From Xinjiang University |
| 红旗镇黑木耳种植合作社 (hóng qí zhèn hēi mù ěr zhòng zhí hé zuò shè) | Qizil bayraq baziri qara mor terish hEmkarliq kopratiwi | Hongqi Town Black Fungus Planting Cooperative |
| 阿勒泰地区文化体育广播电视和旅游局 (a lè tài dì qū wén huà tǐ yù guǎng bō diàn shì hé l ǚ yóu jú) | Altay wilayEtlik mEdEnyEt tEntErbiyE radiyo telewiziyE wE sayahEt idarisi | Aletai Regional Bureau of Culture, Sports, Radio, Television and Tourism |

The Chinese language developed as an isolated language, while the Uyghur language is a morphologically rich agglutinative language. Uyghur words also express a grammatical function, with an inflected form of nouns and verbs. For example, the Uyghur ON, "shinjang dashö" (meaning "Xinjiang University"), is the non-inflected form of the inflected "shinjang dashödE" (meaning "at Xinjiang University"), where the noun suffix, "dE" (meaning "at"), is appended to the end of the word, "dashö" (meaning "university"). This type of inflection not only occurs for the last word in the ON but also for the middle portion. For example, in Table 1, the Uyghur word, "wilayEtlik" (meaning "regional"), is another type of inflection for the initial uninflected "wilayEt" (meaning "region"). The structures of ONs are complex and often nested, leading to a challenging sub-task for named entity translation.

### 3.2. Word Segmenting

The input unit granularity is directly correlated with vocabulary size and has a significant influence on the NMT performance. Word-based neural network models often encounter limited vocabulary size, rare word and out-of-vocabulary (OOV) problems. OOV problems are especially common in the Uyghur language due to its rich morphological nature. As such, we proposed a different word segmentation approach for ON translation tasks, involving characterization, tagged characterization, byte pair encoding (BPE) and syllabification, in order to overcome the vocabulary size, rare-word and OOV problems. These word segmentation approaches will be introduced in the following sub-section.

Statistical studies on language granularity have suggested that Chinese organization names are typically condensed into 2–5 words, whereas Uyghur organization names mostly range from 3–8 words (see Figure 1). As shown in Figure 1, there is an obvious difference between the Chinese and Uyghur ON character length distributions. Most of the Chinese ON samples are 3–15 characters long, while the Uyghur names range from 20–80 characters in length. Table 2 demonstrates the proposed segmentation approaches, with the example of "德拉蒙德公司" (CPA: "Dé lā méng dé gōng sī," meaning "Drummond Corp"), which has the Uyghur translation of "delamond shirkti."

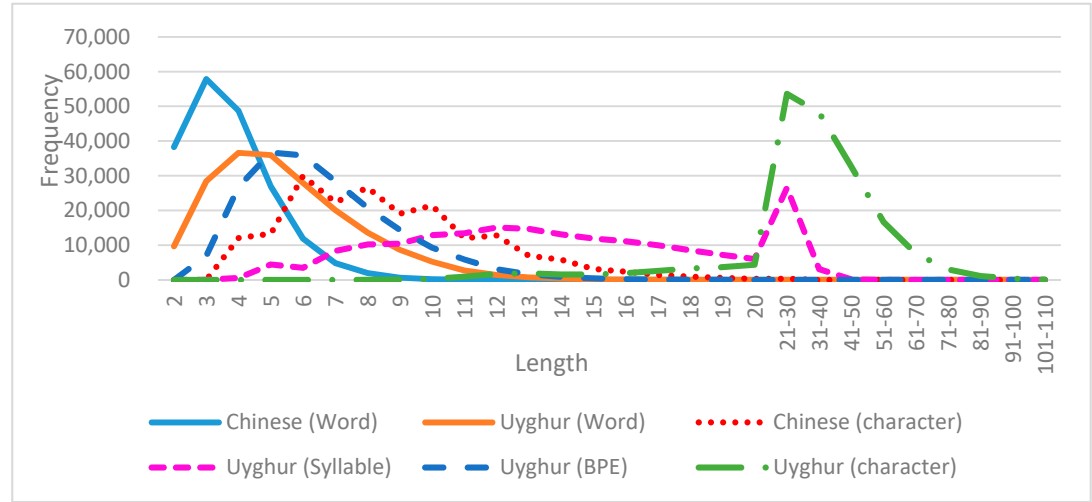

**Figure 1.** A comparison of Chinese and Uyghur organization name lengths with varying granularities.

**Table 2.** Examples of Chinese and Uyghur organization name (ON) word segmentation with varying granularities. BPE: Byte pair encoding.

| Segment Granularity | Chinese | Uyghur (Latin Script) |
|---|---|---|
| Word | 德拉蒙德公司 | delamond shirkti |
| Character | 德拉蒙德公司 | d e l a m o n d s h i r k t i |
| Tagged Character | - | d_B e_M l_M a_M m_M o_M n_M d_E s_B h_M i_M r_M k_M t_M i_E |
| BPE | 德拉@@ 蒙@@ 德公司 | de@@ lam@@ on@@ d shirkti |
| Syllable | - | de la mond shir kti |

### 3.2.1. Characterization and Tagged Characterization

Characters are the basic unit used in composing words. While there are more than 100,000 Chinese characters, only a few thousand are used daily and the exact number is unknown. According to statistical samples, the 1000 most commonly used Chinese characters cover ~92% of written material, 2000 characters cover more than 98% and 3000 characters cover 99% [25].

Uyghur is a morphologically rich agglutinative language, whose current alphabet contains 32 characters [26], including 24 consonants, namely, "b, p, t, j, ch, h, d, r, z, c, s, sh, gh, f, q, k, g, ng, l, m, n, h, w and y" and 8 vowels, namely, "a, E, e, i, o, ö, u and ü."

The Chinese–Uyghur organization name translation corpus contains 3006 Chinese characters, 32 Uyghur characters and 55 symbols. The 55 symbols include Arabic and Roman numbers, some English letters and some punctuations.

Due to differences in character counts, the original Chinese characters were used in the experiment and the Uyghur characters were used as untagged and tagged patterns, respectively. In this work, we propose a new word characterization approach, called tagged characterization and a tagging scheme was implemented in which "B," "M" and "E" were used to indicate the beginning, middle and ending characters in each word, respectively. We append "_" to every character in one of the three tags according to the position of the character in the word, as shown in Table 2.

### 3.2.2. Byte Pair Encoding

Out-of-vocabulary (OOV) issues in NMT have previously been addressed using a sub-word segmentation model, called byte pair encoding (BPE) [27]. BPE is a widely used tokenization technique with obvious advantages in terms of overcoming OOV problems. With this process, words can be

segmented into one or more sub-word components, in which the length of the input or output ON after BPE is subsequently shorter. This approach iteratively merges the most frequently occurring characters or character sequences. In this study, an open source sub-word NMT toolkit [28] was used for both the source input Chinese ON sequences and target output Uyghur ON sequences.

### 3.2.3. Uyghur Syllabification

Uyghur words consist of combinations of syllables, with no special signs. The inherent syllable structure is "initial sound," then "nucleus sound," then "final sound," where the nucleus sound must be a vowel. There can be no initial sound or final sound in the syllable but there must be a nucleus sound [26,29]. Present-day Uyghur syllables include 12 syllabic types, V, VC, CV, CVC, VCC, CVCC, CCV, CCVC, CCVCC, CVV, CVVC and CCCV, where C represents a consonant and V represents a vowel. Wayit et al. performed syllable segmentation with 713,716 unique words appearing in 52,718 Uyghur news articles [26] and identified a total of 8621 different syllables belonging to the 12 syllabic structures [26,29]. We employ syllabification in this work to enable the deep learning architecture to obtain more language-specific knowledge from the syllables.

### *3.3. Neural Network Frameworks*

The sequence-to-sequence architecture has recently become the most popular machine translation framework. In this study, two NMT architectures—namely, a recurrent neural network (RNN) attention-based framework and the current state-of-the-art framework transformer—are proposed for Chinese–Uyghur and Uyghur–Chinese ON translation with different input granularities.

### 3.3.1. The RNN-Based Attention Framework

One of the primary advantages of NMT, a common technique used for machine translation, is that the model learns all features and weights directly from the data, without artificial feature engineering. This allows for the modeling of long-distance dependencies and complicated alignment relationships during the training process [30].

Given a source sequence $x = (x_1, x_2, \ldots, x_n)$ and its corresponding target sequence $y = (y_1, y_2, \ldots, y_m)$, the encoder uses a bi-directional RNN to encode x into a series of hidden states, concatenating the forward and backward hidden states in each position $h = [\overrightarrow{h_1}:\overleftarrow{h_1}, \overrightarrow{h_2}:\overleftarrow{h_2}, \ldots, \overrightarrow{h_n}:\overleftarrow{h_n}]$. After outputting a target sequence $y_{<t} = y_1, y_2, \ldots, y_{t-1}$, a subsequent prediction $y_t$. is generated during the decoding step $t$. The corresponding probability is given by the following:

$$p(y_t|y_{<t}, x) = softmax\big(f\big(s_t, y_{t-1}, c_t\big)\big), \tag{1}$$

where $f(\cdot)$ is a non-linear activation function and $s_t$ is the hidden decoder state at step $t$. It can be expressed as follows:

$$s_t = g\big(s_{t-1}, y_{t-1}, c_t\big). \,, \tag{2}$$

where $g(\cdot)$ is a non-linear activation function and $c_t$ is a context vector, computed as the weighted sum of the encoder hidden states. It can be represented as follows:

$$c_t = \sum_{j=1}^{n} \alpha_{t,j} h_j, \tag{3}$$

where $h_j$ is a hidden state of the source input $x_j$, with a corresponding weight $\alpha_{t,j}$, which can be calculated using the attention model. The target sequence can be factorized as follows:

$$p(y|x, \theta) = \prod_{t=1}^{m} p(y_t|y_{1:t-1}, x, \theta), \tag{4}$$

where $\theta$ denotes the trained model parameters. The training set $P$ in this process consists of parallel sentence pairs (x, y) and the NMT model employs an encoder–decoder framework to jointly train and minimize a negative log-likelihood loss function, given as follows:

$$-\theta = \sum_{(x,y)\in P} -\log p(\mathrm{y}|x,\theta).$$ (5)

In this study, Google's neural machine translation (GNMT) model was adopted for RNN-based attention model training, as shown in Reference [31].

### 3.3.2. The Transformer

The transformer framework is the current state-of-the-art architecture for NMT. When used in References [32,33], the framework outperformed a comparative RNN-based attention model [30]. The framework is unique, as both the encoder and decoder are composed of stacked self-attention and point-wise fully connected layers, without any explicit recurrent structure. The encoder is composed of a stack of N identical layers, each consisting of a multi-head self-attention mechanism sub-layer and a position-wise fully connected feed-forward network (see Figure 2).

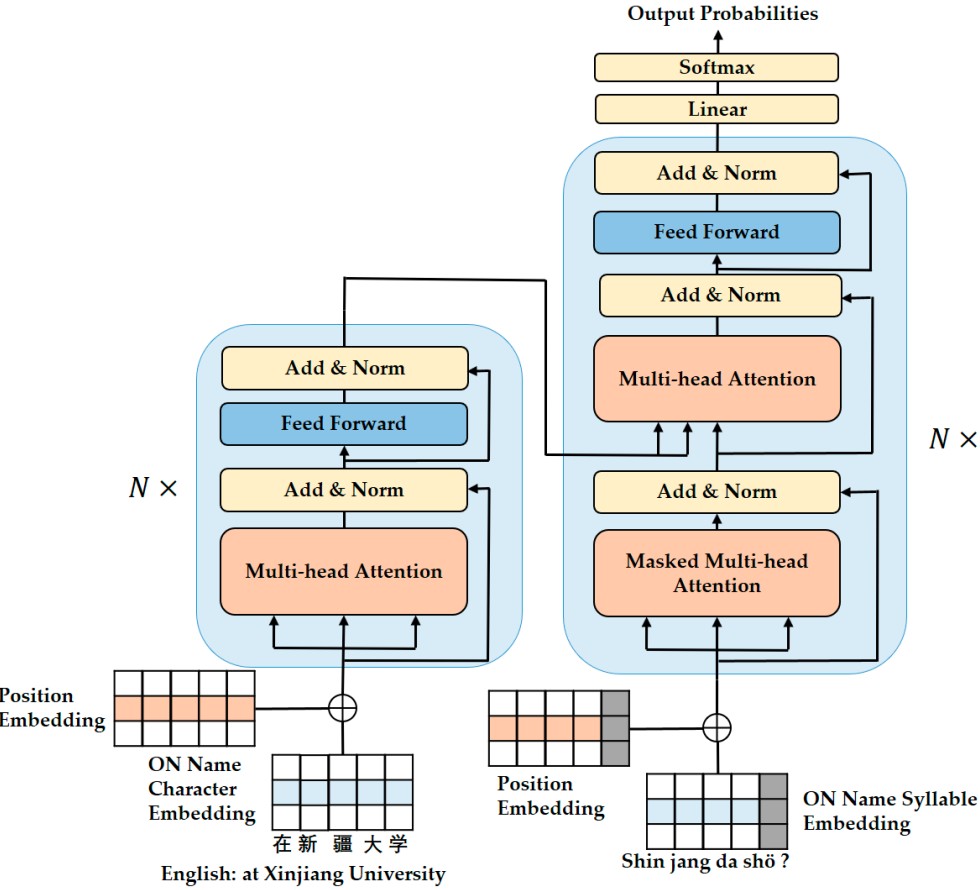

**Figure 2.** Transformer architecture. This figure illustrates the character-level Chinese input as the source language and the syllable-level Uyghur input as the target language. The input Chinese sequence is "在新疆大学" (CPA: "zài xīn jiāng dà xué," meaning "at Xinjiang University") and "shin jang da shö" is the target predicted syllable sequence. The gray column of the target input vectors indicates a masked vector, which has not yet been predicted.

Similarly, the decoder also includes a stack of N identical layers and two sub-layers; however, it also has an additional multi-head attention sub-layer that performs multi-head attention operations

with the encoder output. This transformer can be used to modify the self-attention sub-layers in the decoder stack and prevent positions from occupying subsequent positions. This masking operation, combined with the output embedding offset by a single step, ensures that the prediction $i$ depends only on the known outputs at positions less than $i$. In contrast, conventional algorithms often use residual connections around each of the two sub-layers, followed by layer normalization [32].

The encoder can be used to transform an input sequence $x = (x_1, x_2, \ldots, x_n)$ into a sequence of continuous hidden representations $z = (z_1, z_2, \ldots, z_n)$, from which the decoder generates a target sequence $y = (y_1, y_2, \ldots, y_m)$, one element at a time. Vaswani et al. proposed a multi-head self-attention mechanism that allows this model to jointly attend to information from different representation subspaces at different locations through the use of a scaled dot product as the alignment score [25]. This process can be described as mapping a query and a set of key–value pairs to an output, operating on a query $Q$, a key $K$ and a value $V$ as follows:

$$\text{Attention}(Q, K, V) = softmax\left(\frac{QK^T}{\sqrt{d_k}}\right)V, \tag{6}$$

where $d_k$ is the dimension of the key.

The terms $Q$, $K$ and $V$ are vector representations of input sequence symbols for the first encoder and decoder. Each of these terms represents the output of a hidden-state matrix in the previous layer for multi-head intra-attention mechanisms. Specifically, $Q$ denotes the hidden states in the previous decoder layer, in which $K$–$V$ pairs can be generated from the output $(z_1, z_2, \ldots, z_n)$ of the encoder. Multi-head attention can then be used to acquire $h$ different representations of $(Q_i, K_i, V_i)$. The transformer is used to project a hidden-state matrix into a distinct query, key and value representation $(Q_i = QW_i^Q, K_i = KW_i^K, V_i = VW_i^V)$ for each attention head $i$. Scaled dot product attention is then performed for each representation, the outputs are concatenated and the results are projected using a point-wise fully connected feed-forward network as follows:

$$\text{MultiHead}(Q, K, V) = Concat(head_1, \ldots, head_n)W^o, \tag{7}$$

$$head_i = \text{Attention}\left(QW_i^Q, KW_i^K, VW_i^V\right), \tag{8}$$

where $W_i^Q, W_i^K, W_i^V$ and $W^o$. are parameter projection matrices.

### 3.4. Chinese–Uyghur ON Translation Pair Generating System

The size of a bilingual ON corpus has a significant effect on model performance in applications, such as machine translation or cross-lingual information processing; however, artificial ON translation, which is required to develop and expand corpora, is time-consuming and tedious. As such, a round-trip Chinese–Uyghur ON translation pair generation system was developed as part of this study to automatically extend a corpus. The quality of these entries was ensured using Chinese–Uyghur ON translations and an Uyghur–Chinese ON model, as shown in Figure 3.

The Chinese NE recognizer extracts Chinese organization names (CONs) from Chinese texts and the system retains entries that are not already in the Chinese–Uyghur ON pair corpus, thereby forming a CON translation dataset, denoted as $CON = \{CON_1, \ldots, CON_n\}$. The Chinese–Uyghur ON translation model is then used to translate Chinese–Uyghur ONs from the CON dataset into Uyghur organization name ($UON$) data, denoted as $UON = \{UON_1, \ldots, UON_n\}$. The Uyghur–Chinese ON translation model then translates $UON = \{UON_1, \ldots, UON_n\}$ back into $\widehat{CON} = \left\{\widehat{CON_1}, \ldots, \widehat{CON_n}\right\}$, where the hat symbols indicate synthetic CONs. Finally, the system inserts $CON_i$ and $UON_i$ into the Chinese–Uyghur ON pair corpus, if the $CON_i$ and $\widehat{CON_i}$ terms are identical.

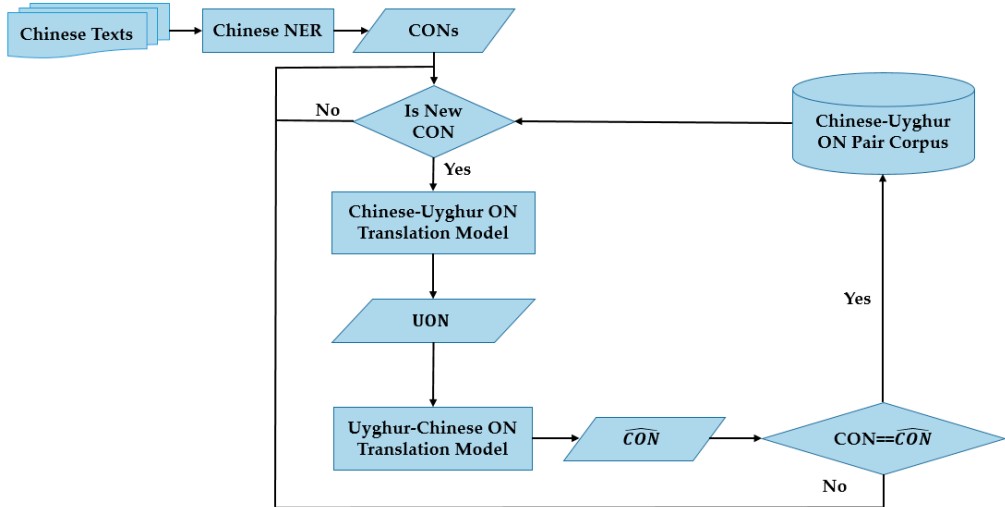

**Figure 3.** A flow chart for the proposed Chinese–Uyghur ON translation pair generation system.

## 4. Experimental Results and Discussion

### 4.1. Chinese–Uyghur Organization Name Translation Corpus Construction

There are no publicly available Chinese–Uyghur NE corpora at present, aside from the CCMT (China Conference on Machine Translation) dataset, which contains limited training (6318 ONs) and testing samples (143 ONs). As such, one of the primary contributions of this study is the development of a Chinese–Uyghur ON translation corpus for future research. In this process, text from websites published by the People's Daily Online, Xinhua Net, China Central Television (CCTV) and State Council was extracted using a web crawler. Chinese named entities were then isolated using a language technology platform (LTP) [34]. Incorrect ON labels were filtered out manually and the remaining terms were translated from Chinese to Uyghur using an NMT. The synthesized Uyghur ON terms were then translated back into Chinese using a Uyghur–Chinese NMT. Named pairs were kept if their initial Chinese ON and round-trip translated Chinese ON were in agreement (~10% of the data were in agreement). This produced 191,641 pairs of Chinese–Uyghur ON translations (See Supplementary Materials). Statistical studies have suggested the existence of 6373 different Chinese ON suffixes. The most frequent 15 ONs accounted for more than 90% of these data (see Figure 4).

The coverage of ONs in the data was relatively comprehensive and the quality was relatively high. As such, we divided the samples into training, development and test sets, as shown in Table 3.

**Table 3.** The experimental dataset.

| Dataset | Size |
|---------|---------|
| train | 181,924 |
| dev | 2000 |
| test | 7717 |

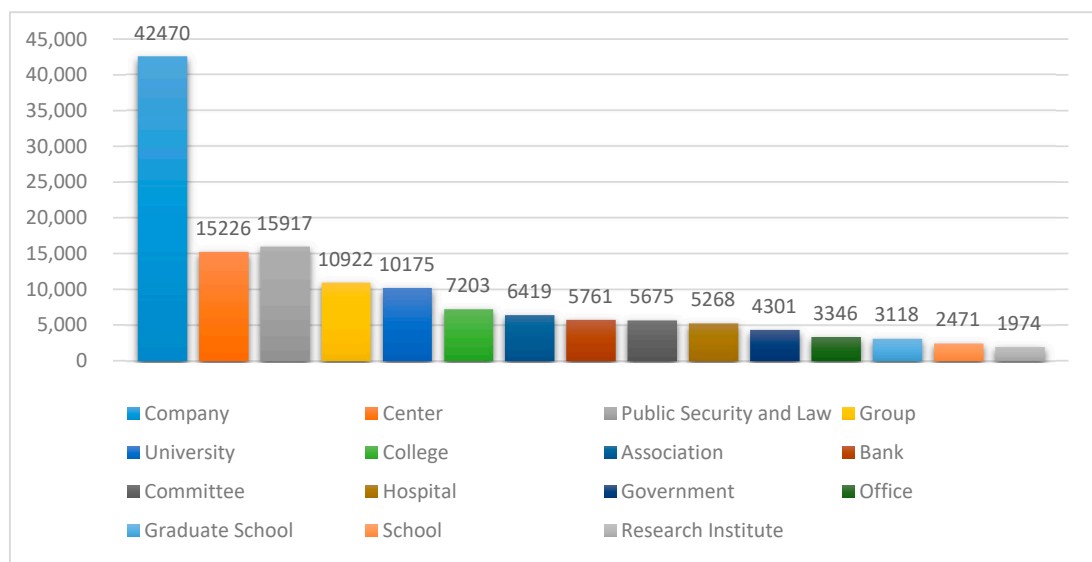

**Figure 4.** Diagram of the top 15 organization name suffixes, representing business areas and the organization categories.

### 4.2. Evaluation Metrics

In this study, accuracy (ACC) and BLEU scores were used to evaluate the resulting translation model. Accuracy measures the percentage of correctly translated ON terms, in which the top candidate ON produced by the translation model identically agrees with the ground truth ON. This can be expressed mathematically as follows:

$$ACC = \frac{\text{Number of translated ON terms matching the ground truth}}{\text{Total ON count}}. \tag{9}$$

The bilingual evaluation understudy (BLEU) [35] metric is a standard assessment metric used to compare machine translation models with manually-produced results. BLEU was used in this study to match variable length phrases with machine output and reference translations. It is defined as follows:

$$BLEU = BP * exp\left(\sum_{n=1}^{N} w_n * \log P_n\right), . \tag{10}$$

where $w_n$ are positive weights, which sum to 1. The n-gram precision $P_n$ can be calculated using $n$-grams, with a maximum length of $N$. $BP$ is a brevity penalty, calculated as follows:

$$BP = \begin{cases} 1, & if\ h > r \\ exp^{1-\frac{h}{r}}, & if\ h \leq r \end{cases}. \tag{11}$$

Weighted matched averages can also be used to determine a translation score.

### 4.3. Experimental Settings

This section introduces the experimental configuration, including details concerning the open source toolkit and model hyperparameters. Here, the ON translation models were trained separately, with varying input granularities (i.e., characters), BPEs, words (for both the source and target languages), tagged characters and syllables (only for Uyghur ONs).

1.  GNMT [36] was used as an RNN-based attention training system. The dimensions of both the input and output sequence vectors, with different granularities, as well as the hidden state, were

set to 512. Four hidden layers were used in both the encoder and decoder and a stochastic gradient descent (SGD) algorithm was included as an optimizer. The learning rate was set to 1.0 and the dropout rate was 0.2. The decoder used beam search decoding, with a beam size of 3 and a batch size of 128 and the default settings were used for all other hyperparameters.

2.  OpenNMT [37], an open-source ecosystem for neural machine translation and neural sequence learning, was used for transformer-based model training. Six encoder and decoder layers were included, in which both the input and output sequence vector dimensions were set to 512. The encoder and decoder included 16 attention heads, the feed-forward inner-layer dimension was set to 4096 and the dropout rate was set to 0.3. The Adam algorithm was included as an optimizer, with adam_beta1 = 0.9 and adam_beta2 = 0.998 and the default settings were used for all other hyperparameters.

3.  The open-source sub-word NMT toolkit [28], with a vocabulary size of 16,000 for both Chinese and Uyghur, was used for BPE. NLTK [38] was used to calculate the BLEU scores (see Table 4).

**Table 4.** Statistics for the training dataset, with varying granularities.

| Granularity | Chinese | Uyghur |
|---|---|---|
| Word | 39,554 | 25,509 |
| Character | 3007 | 87 (55 symbols) |
| Uyghur tagged character | - | 283 |
| Uyghur syllable | - | 6995 |

### 4.4. Results and Discussion for Chinese–Uyghur ON Translation

A series of experiments were conducted to compare different neural machine models and varying input unit granularities in order to determine the optimal conditions for Chinese–Uyghur ON translation. The Seq2Seq attention model and transformer were assessed and compared with different inputs, such as characters, tagged characters, BPEs, syllables and words. The BPE segmentation toolkit [28] was used to segment Chinese and Uyghur phrases, while the segmentation algorithm developed by Wayit et al. [29] was used for Uyghur syllable segmentation. The sample data used for training and testing are shown in Table 3 and the quantitative results are provided in Table 5, where 'char' stands for 'character.'

**Table 5.** Chinese–Uyghur ON translation results for varying input units. BLEU: Bilingual evaluation understudy. GNMT: Google's neural machine translation.

| Input Granularity | Method | BLEU | | Accuracy (%) | |
|---|---|---|---|---|---|
| | | Dev | Test | Dev | Test |
| Char–Char | GNMT | 47.34 | 43.46 | 32.20 | 27.00 |
| | Transformer | 81.78 | 80.74 | 71.40 | 69.83 |
| Char–TaggedChar | GNMT | 45.67 | 42.05 | 33.80 | 28.78 |
| | Transformer | 73.49 | 71.64 | 63.95 | 60.87 |
| BPE–BPE | GNMT | 79.01 | 77.72 | 66.80 | 65.06 |
| | Transformer | 84.16 | 84.35 | 74.30 | 74.67 |
| BPE–Syllable | GNMT | 75.87 | 74.91 | 63.20 | 61.90 |
| | Transformer | 84.30 | 83.69 | 74.50 | 73.75 |
| Char–Syllable | GNMT | 76.67 | 75.47 | 64.15 | 62.01 |
| | Transformer | 85.06 | 85.17 | 74.75 | 74.71 |
| Word–Word | GNMT | 70.98 | 70.88 | 55.90 | 56.20 |
| | Transformer | 79.96 | 79.59 | 65.90 | 66.72 |

These experimental results indicate that the transformer outperformed the RNN attention-based GNMT system for Chinese–Uyghur ON translation. The maximum and minimum differences in the BLEU scores and accuracy values were 34% and 42% and 5.15% and 11.1%, respectively.

It can clearly be seen that there were obvious performance differences between the various input granularities (see Figure 5). The discrepancies between the word-based models and the Char–Syllable algorithm demonstrate the importance of syllabification for Uyghur ON. There were few differences between BPE–BPE, BPE–Syllable and Char–Syllable. This suggests that a sub-word unit input for Uyghur produced better results for Chinese–Uyghur ON translation.

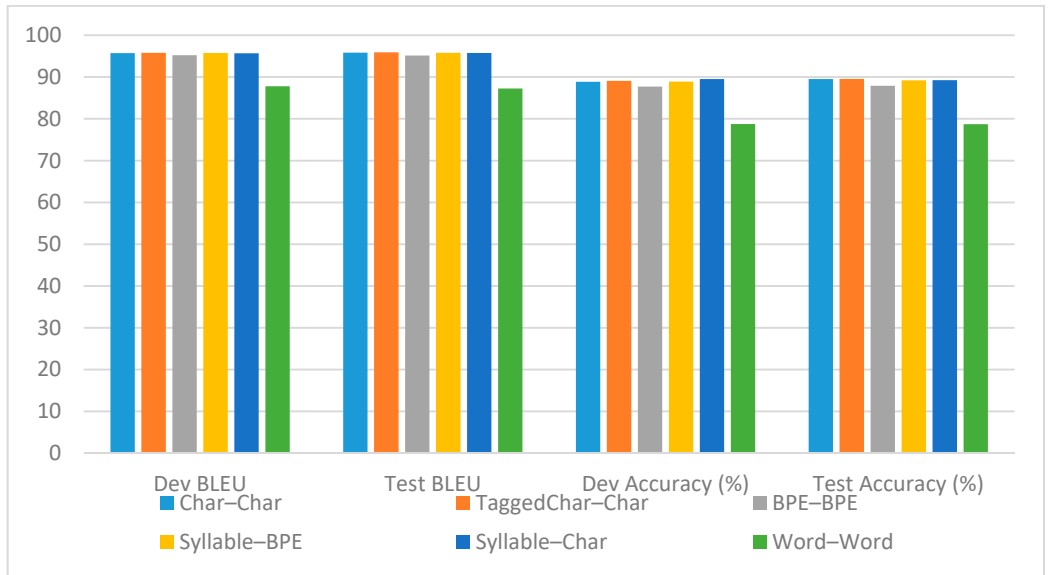

**Figure 5.** Comparison of the transformer-based Chinese–Uyghur ON translation model with varying segment granularities.

By analyzing translation results from the test set, we found that the majority of errors could be classified into one of six categories, as listed in Table 6. Character insertion represents the most frequent error in Chinese–Uyghur ON translation tasks due to the agglutinative nature of the Uyghur language. For example, the suffixes, "lik" and "i, si," are often appended to words to express affiliation. The term, "shEhEr," means "city" but "shEhErlik" means "municipal." As such, "shEhElik hvkvmet" means "municipal government." The suffixes, "i, si and lik," always appear at the end of words (particularly "i"), making these mistakes more common. Errors also produced large differences between the BLEU and accuracy scores in the experimental validation process. The BLEU values were calculated using n-gram matching, in which the insertion or absence of one or two characters did not make a significant difference. In contrast, the accuracy score requires an entire ON to match the ground truth exactly. For example, "filorida ishtati hokvmiti" differed from the ground truth, "filorida shitati hokvmiti," by only a single character and "xuEy'En shEhiri muhit asrash idarisi" (meaning "Huaian Environmental Protection Bureau") differed from "xuEy'En shEhErlik muhit asrash idarisi" by a single word. These two ON translations, therefore, improved the BLEU score but decreased the accuracy.

**Table 6.** Error categorization and examples of the best Chinese–Uyghur translation model.

| Error Category | Error ON (Latin Script) | Correct ON (Latin Script) | English |
|---|---|---|---|
| Character insertion | Filorida ishtati hokvmiti | Filorida shitati hokvmiti | Florida Government |
| Character missing | XuEy'En shEhiri muhit asrash idarisi | XuEy'En shEhErlik muhit asrash idarisi | Huaian Environmental Protection Bureau |
| Synonym replacement | Xuayi qërindashlar mEdEniyEt tarqitish pay chEklik shirkiti | Xuayi aka—uka mEdEniyEt tarqitish pay chEklik shirkiti | Huayi Brothers Media Group |
| Word insertion | Da guangming soda chEklik mEsuliyEt shirkiti | Da guangming soda chEklik shirkiti | Daguangming Commerce co., Ltd. |
| Wrong translation | Kongo altun hvkvmeti | Kongo kinsasha hvkvmeti | Kongo Kinsasha Goverment |
| Polyphonic translation | Sheriq desey mEktipi | Dongfang desey mEktipi | Dongfang Decai School |

*4.5. Results and Discussion for Uyghur–Chinese ON Translation*

Experiments in the Uyghur–Chinese translation direction were conducted with the same settings as the Chinese–Uyghur ON translation task and the results are shown in Table 7.

**Table 7.** Uyghur–Chinese ON translation results with varying input units.

| Input Unit | Method | BLEU | | Accuracy | |
|---|---|---|---|---|---|
| | | Dev | Test | Dev | Test |
| Char–Char | GNMT | 91.64 | 91.41 | 80.02 | 80.01 |
| | Transformer | 95.73 | 95.83 | 88.85 | 89.51 |
| TaggedChar–Char | GNMT | 93.34 | 92.97 | 83.85 | 83.16 |
| | Transformer | 95.78 | 95.90 | 89.10 | 89.56 |
| BPE–BPE | GNMT | 87.50 | 87.45 | 74.55 | 74.12 |
| | Transformer | 95.22 | 95.13 | 87.70 | 87.91 |
| Syllable–BPE | GNMT | 85.14 | 84.90 | 65.20 | 65.60 |
| | Transformer | 95.75 | 95.79 | 88.90 | 89.22 |
| Syllable–Char | GNMT | 88.28 | 87.68 | 75.10 | 74.73 |
| | Transformer | 95.69 | 95.74 | 89.50 | 89.24 |
| Word–Word | GNMT | 82.72 | 83.7 | 66.75 | 67.12 |
| | Transformer | 87.80 | 87.27 | 78.75 | 78.72 |

The experimental results shown in Table 7 indicate that the transformer still outperformed the RNN attention-based GNMT system for Uyghur–Chinese ON translation. It can be seen, from Table 7 and Figure 6, that the Uyghur–Chinese ON translation approach based on the transformer also benefited from the characterization and sub-word segmentation. The tagged Uyghur and untagged Chinese character-based model achieved the best result, without a significant difference, compared to other models (except the word-based model). All of the character-, tagged character-, BPE- and syllable-level-based models surpassed the word-based model, with an accuracy increase greater than 10% and an increase in BLEU points of 7.98.

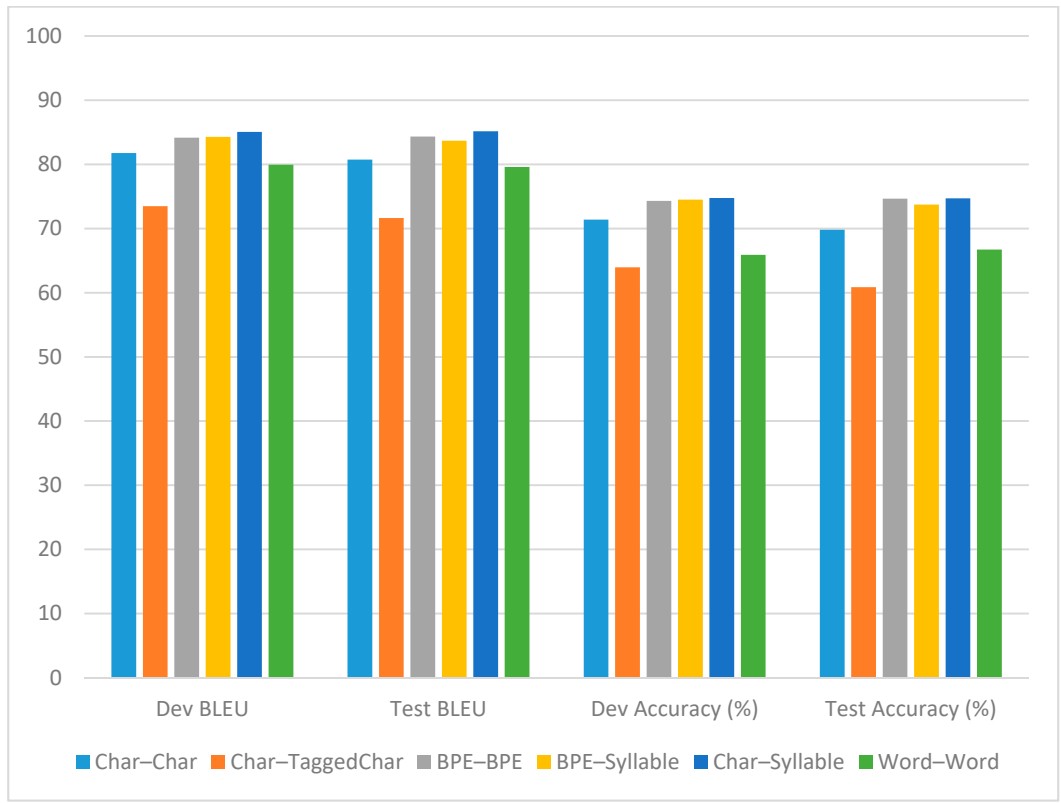

**Figure 6.** Comparison of the transformer-based Uyghur–Chinese ON translation model with varying segment granularities.

Frequent translation errors made by the best translation model were classified to produce the eight categories shown in Table 8. Homophone errors were the most common due to the frequency of transliteration for ONs. However, Uyghur is an alphabetical language and does not exhibit a homophonic structure. For example, the two Chinese words, "马里基" (CPA: "mǎ lǐ jī," meaning "Maliki") and "马利基" (CPA: "mǎ lì jī," meaning "Maliki"), have exactly the same pronunciation and form in the Uyghur language ("Maliji"). Character insertion is another common error, caused by attempts to clarify ON meaning but preventing results from agreeing with a reference (see Table 8). This became problematic, as some abbreviations only appeared in the source or target language, which confused the model. For example, "社科院" (CPA: "shè kē yuan," meaning "Academy of Social Sciences") is an abbreviation of "社会科学院" (CPA: "shè huì kē xué yuan," meaning "Academy of Social Sciences") and has no abbreviation in Uyghur. Synonym replacement was another common error. For example, "qërindashlar" (meaning "brothers") and "aka-uka" (meaning "older and younger brothers") express similar meanings. In addition, some words should be translated semantically, while others should be transliterated. For example, "东" (CPA: "dōng," meaning "east") can be translated semantically as "sheriq" (meaning "east") but is transliterated as "dong" when using phonemes in some contexts. The translation of "东北大学" (CPA: "dōng běi dà xué," meaning "northeastern university") is "sheriq shimal daxuesi" but the translation of "华东公司" (CPA: "huá dōng gōng sī," meaning "Huadong Company") is "huadong sherkiti."

**Table 8.** Error categorization of the best Uyghur–Chinese translation model. CPA: Chinese phonetic alphabet.

| Error Category | Error ON (CPA) | Correct ON (CPA) | English |
|---|---|---|---|
| Character insertion | 上海网络游戏公司(shàng hǎi wǎng luò yóu xì gōng sī) | 上海网游公司(shàng hǎi wǎng yóu gōng sī) | Shanghai Online Game Company |
| Character missing | 中沙高级联合委员会(zhōng shā gāo jí lián hé wěi yuán huì) | 中沙高级别联合委员会(zhōng shā gāo jí bié lián hé wěi yuán huì) | China-Saudi Arabia High-level Joint Committee |
| Synonym | 雪歌服装有限公司(xuě gē fú zhuāng yǒu xiàn gōng sī) | 雪歌服饰有限公司(xuě gē fú shì yǒu xiàn gōng sī) | Xuege Clothing Co., Ltd. |
| Word insertion | 美国公共选举竞争基金会(měi guó gōng gòng xuǎn jǔ jìng zhēng jī jīn huì) | 美国公共竞选基金会(měi guó gōng gòng jìng xuǎn jī jīn huì) | American Public Campaign Foundation |
| Word translation | 昆明报价中心(kūn míng bào jià zhōng xīn) | 昆明价格举报中心(kūn míng jià gé jǔ bào zhōng xīn) | Kunming Price Report Center |
| Abbreviation | 北京社会科学院经济研究所(běi jīng shè huì kē xué yuàn jīng jì yán jiū suǒ) | 北京社科院经济研究所(běi jīng shè kē yuàn jīng jì yán jiū suǒ) | Institute of Economics of Beijing Academy of Social Sciences |
| Homophone | 马里基政府(mǎ lǐ jī zhèng fǔ) | 马利基政府(mǎ lì jī zhèng fǔ) | Maliki Government |

*4.6. Results and Discussion for the Chinese–Uyghur Organization Name Translation System*

The accuracy values of the Char–Syllable and Char–Char (tagged) Chinese–Uyghur ON translation models reached ~74% and ~89%, respectively. The high quality of these results prompted the development of a Chinese–Uyghur ON translation pair generation system. This system was tested using the CCMT Chinese monolingual corpus, which includes 700 million sentences. The LTP extracted 265,930 candidate CONs, 41,691 of which already existed in the corpus. After filtering 224,239 terms, 180,822 CONs remained. After round-trip translation, 102,626 Chinese–Uyghur ON translation pairs were identified, representing 56.76% of the initial 180,822 CONs, as shown in Table 9. A Chinese–Uyghur NMT system was then used to construct the translation pair corpus, comprising ~10% of the original data. This result demonstrates the effectiveness of the proposed ON translation models.

**Table 9.** Test procedure statistics.

| Processing Step | Count |
|---|---|
| CON recognized by LTP | 265,930 |
| CON not in corpus | 224,239 |
| After filtering | 180,822 |
| Final ON translation pair | 102,626 |

## 5. Conclusions

In this study, a novel named entity translation approach was introduced for Chinese–Uyghur organization name dataset translation and pair construction. As there is currently no publicly available NE translation corpus, a Chinese–Uyghur ON dataset was constructed as part of this study. These data consisted of ~200,000 Chinese–Uyghur ON pairs, which may be used for further research on and improvement of Chinese–Uyghur NMT machine translation and cross-lingual NE recognition systems. A series of word segmentation approaches were proposed in this work in order to explore the impact of various input granularities on Chinese–Uyghur machine translation tasks. Two main neural machine translation frameworks were adopted for organization name translation tasks, in which the transformer framework outperformed the other model in both translation directions. The experimental results also indicated that the organization name segmentation results found by the characterization and sub-word units (i.e., syllables, BPEs and tagged characters) not only mitigated unknown input units but also improved the model accuracy in both translation directions. A round-trip

translation-based Chinese–Uyghur ON generation system was further developed to extend the corpus automatically. The results of a series of validation tests demonstrated the effectiveness of the ON translation models proposed in this work. As the Uyghur–Chinese machine translation competition test data only included 143 ONs—which is almost negligible for standard data—the model was not integrated into an NMT system (a common testing practice). In a future study, we will further expand the corpus size using the proposed model and investigate the filtering of misrecognized ONs with bilingual information. We also intend to study mistranslations, synonym errors and the automatic evaluation of confidence using round-trip translation and variations in language models.

**Supplementary Materials:** The following are available online at https://github.com/hasan1479/ON_name.

**Author Contributions:** Conceptualization, A.W. and C.X.; methodology, A.W., C.X. and Z.K.; software, C.X., X.H. and W.L.; validation, C.X., Z.K. and Y.W.; formal analysis, A.W.; resources, C.X., A.S. and M.M.; writing—original draft preparation, A.W. and C.X.; writing—review and editing, A.W. and S.T.; supervision, A.W.; funding acquisition, A.W. All authors have read and agreed to the published version of the manuscript.

**Funding:** This study was funded by the Opening Foundation of the Key Laboratory of Xinjiang Uyghur Autonomous Region of China (grant no. 2016D03023); the National Natural Science Foundation of China (grant no. 61662077); the Scientific Research Program of the State Language Commission of China (grant no. ZDI135-54); and the National Key Research and Development Project of China (grant number 2017YFB1002103).

**Conflicts of Interest:** The authors declare no conflict of interest.

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
