# Peer review of "A Neural-Network-Based Approach to Chinese–Uyghur Organization Name Translation"

_information, doi:10.3390/info11100492_

Round 1

Reviewer 1 Report

This paper aims to help improve Chinese-Uyghur translation of organization names (ON). The authors introduce a novel translation system capable of automatic generating new Chinese-Uyghur ON translation pairs. Their findings may be applied not only to Chinese-Uyghur translations but also to Chinese-to-other-agglutinative-languages ON translations.

Although I do not understand Chinese or Uyghur, I appreciate the author´s effort to improve ON machine translation. I presume that this area of machine translation is particularly difficult to deal with. The choice of topic is good, the background information including references is sufficient. The authors also describe potential pitfalls posed by the fact that Chinese and Uyghur used different characters (Chinese logograms versus Latin script). The research shows that the transformer framework is a better option. Its architecture is depicted in Figure 2 and described in the text. The authors also developed a translation pair generation system to extend the Chinese-Uyghur ON translation corpu. The way of diong it is also described in the text.

The text as a whole is written in good English, it is accompanied by both tables and figures that help the reader take more from the text.

Nonetheless, I recommend the authors to do proofreading for minor slips.

Author Response

Dear Reviewer,
I revised my manuscript carefully and submitted for English Editing to the MDPI English Editing Service. The resubmitted manscript war proofreaded. I also provided the English Editing Certificate as an supplentary.
Yours Sincerely
The Author

Reviewer 2 Report

The paper focuses on a very specific Natural Language Processing topic, namely the correct Chinese–
Uyghur translation of Organization Names. 

The topic of the paper can be considered very narrow, but nevertheless relevant. The work could be relevant for improving translations between languages, when a large translation corpus is not available. The authors have created a dedicated corpus for the proposed topic. The paper is well written and can be considered technically sound. 

# Recommendations

- In order to facilitate the reproducibility of the results and to promote further research the authors should either include the developed corpus as supplementary material to the paper or upload it to a public repository such as Github and include the link towards the repository in the paper.

- At the beginning of section 3.2. and in Table 2 the paper mentions the concept of "tagged characterization", however the concept is only explained towards the end of sub-section 3.2.1. The authors are kindly asked to include at least a brief explanation of the concept when mentioning it for the first time (or to specify that the concept will be explained in the following sub-section).

- Axes labels should be added to figure 1.

- In the section 3.2.1, the paper mentions that the developed corpus includes "55 symbols, respectively". The authors are kindly asked to explain what symbols are included in the corpus.

Author Response

Dear Reviewer,
I really appreciate your recommendations, then they help me to improve this manuscript. I revised my manuscript carefully according to your recommendations.
Firstly, I uploaded the whole training and testing data to Github as you recommend. The data is now available at https://github.com/hasan1479/ON_name. I add this to supplementary part of this manuscript.
Secondly, I specified that all the word segmentation approaches will be introduced in the following sub-section at the end of the first paragraph of section 3.2. And I also described the concept of “tagged characterization” more clearly than before.
Thirdly, I added Axes labels to figure 1.
Finally, I added the following sentence to explain those "55 symbols ". The added sentence: “The 55 symbols include Arabic and Roman numbers, some English letters, and some punctuations.” I also uploaded those 55 symbols as a separate file with training and testing files.
I also summited my revised manuscript to the MDPI English Editing Service for English checking and proofreading.
I hope my amendment can meet your suggestion.

Yours Sincerely
The Author

This manuscript is a resubmission of an earlier submission. The following is a list of the peer review reports and author responses from that submission.